# Federated Timeline Synthesis: Scalable and Private Methodology For Model Training and Deployment

## Abstract

We present Federated Timeline Synthesis (FTS), a novel framework for training generative foundation models across distributed timeseries data applied to electronic health records (EHR). At its core, FTS represents patient history as tokenized Patient Health Timelines (PHTs), language-agnostic sequences encoding temporal, categorical, and continuous clinical information. Each institution trains an autoregressive transformer on its local PHTs and transmits only model weights to a central server. The server uses the generators to synthesize a large corpus of trajectories and train a Global Generator (GG), enabling zero-shot inference via Monte Carlo simulation of future PHTs. We evaluate FTS on five clinically meaningful prediction tasks using MIMIC-IV data, showing that models trained on synthetic data generated by GG perform comparably to those trained on real data. FTS has the potential to offer strong privacy guarantees, scalability across institutions, and extensibility to diverse prediction and simulation tasks especially in healthcare, including counterfactual inference, early warning detection, and synthetic trial design. We publish the code at `https://anonymous.4open.science/r/fts-paper`.

## 1 Introduction

Recent breakthroughs in self-supervised learning on large-scale text corpora, most notably the GPT family, have demonstrated the transformative potential of foundation models. However, extending these successes to healthcare presents unique challenges. Beyond strict privacy regulations (e.g., GDPR, CCPA) and data-sovereignty constraints, clinical data is fragmented across institutional silos and marked by substantial heterogeneity. Patient populations vary in demographics, disease prevalence, and progression of medical interventions; documentation practices differ significantly between institutions; and the language used in electronic health records (EHRs) is often domain-specific, inconsistently structured, and highly variable. These factors pose significant obstacles to training centralized, homogeneous foundation models in healthcare. To address these challenges and enable scalable clinical modeling, we introduce *Federated Timeline Synthesis* (FTS), a privacy-preserving, communication-efficient framework for training generative transformers across distributed clinical data. At the core of FTS is a language-agnostic representation of medical information through tokenized *Patient Health Timelines* (PHTs), designed to capture the longitudinal, quantitative, and multimodal structure of real-world healthcare data.

**Patient Health Timelines and Zero-Shot Inference** A patient's longitudinal record can be modeled as an ordered sequence of clinical events, each transformed into one or more discrete tokens per event, analogous to subword tokens in natural language processing (NLP) Kraljevic et al. (2024); Renc et al. (2024); Zhou & Barbieri (2025). To capture the irregular timing of healthcare interactions, *time-interval tokens*, drawn from a predefined set of nominal bins (e.g., 5 min, 20 min, 1 h, ..., 1 yr), can be incorporated into the timeline Renc et al. (2024). Continuous measurements (e.g., laboratory results, vital signs) can be quantized into population-based *quantile tokens*, preserving relative value rankings without revealing exact magnitudes. High-cardinality categorical variables (e.g., ICD codes, medication codes) are tokenized using *hierarchical tokenization*, where each level of the taxonomy contributes one or more tokens, analogous to byte-pair encoding in text. Multimodal inputs such as clinical notes, radiology images, and genomic profiles can be processed through pretrained encoders (e.g., transformers for text, CNNs for images) to produce fixed-dimensional *embedding vectors*,

which are then interleaved into the token sequence. All tokens and embeddings are mapped to a shared continuous vector space, enabling autoregressive transformer architectures (e.g., GPT-style models) to learn longitudinal patterns, temporal dependencies, and intermodal relationships. We refer to this unified, language-agnostic, privacy-aware, temporally resolved tokenized format as the *Patient Health Timeline (PHT)* (Fig. 1). Such a representation enables *zero-shot inference* through Monte

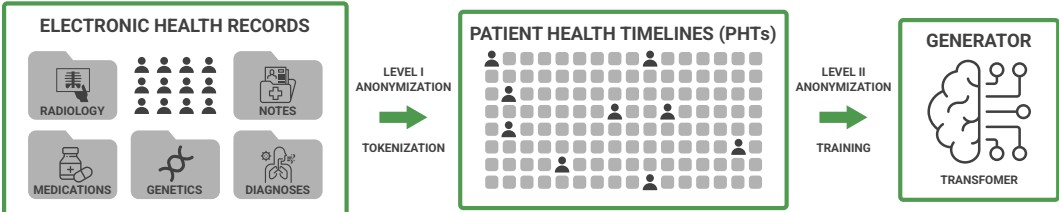

Figure 1: Tokenization and model training introduce two layers of anonymization.

Carlo simulation of future Patient Health Timelines (fPHTs). Given a partial patient timeline and a conditioning prompt (e.g., task-specific token or outcome of interest), the model autoregressively samples multiple plausible future trajectories. Predictions for clinical outcomes are then derived by aggregating statistics over the generated fPHTs, such as the frequency of a target event, the distribution of outcome classes, or the average value of a measurement. This approach allows the model to generalize to previously unseen clinical prediction tasks without requiring task-specific retraining, supporting flexible deployment across diverse clinical settings and outcome types, even in the absence of labeled data (see Sec. 2.1 for details).

**Federated Learning (FL)** Since the introduction of Federated Averaging (FedAvg) by McMahan et al. (2017), which established that local SGD with periodic model averaging can provably converge under non-independent, identically, distributed client distributions, federated learning has rapidly evolved to address five key challenges: statistical heterogeneity, communication efficiency, robustness, personalization, and privacy. There are many innovations in FL developed over the years like proximal-based methods such as FedProx Li et al. (2020a) stabilize client updates via regularization, while control-variate schemes like SCAFFOLD Karimireddy et al. (2020) correct for client drift. For thorough coverage of state of the art refer to Yurdem et al. (2024); Liu et al. (2024); Ji et al. (2024); Choi et al. (2024).

**Synthetic EHR and Federated Synthesis (FS)** Complementary to real-data modeling, synthetic EHR generation has emerged as a strategy to mitigate privacy concerns and data scarcity. Generative models such as *medGAN* Choi et al. (2017) demonstrated the ability to synthesize realistic multi-label patient records. Subsequent methods introduced temporality and multimodality: EHR-M-GAN Baowaly et al. (2019) modeled both continuous and discrete sequences from ICU records, improving utility and realism. Privacy-aware methods such as EHR-Safe Yoon et al. (2023) combine utility and protection against re-identification. Synthetic data from these models have been used to augment predictive tasks, boosting performance and enabling cross-institutional studies Torfi et al. (2022); Theodorou et al. (2023); Zhou & Barbieri (2025). Together, these works highlight the growing role of synthetic data in training and validating clinical foundation models while addressing privacy, fairness, and generalizability constraints.

FS is an emerging paradigm that expands the traditional goals of FL by focusing not only on training shared models but on collaboratively generating synthetic data across distributed clients. Unlike conventional FL, which aggregates model gradients or weights while keeping raw data local, FS aims to produce artificial datasets that approximate the statistical properties of decentralized data without exposing individual records. This synthetic data can then be used for downstream machine learning tasks, simulation, or model validation in privacy-sensitive domains such as healthcare. Most approaches to federated synthesis rely on deep generative models, such as GANs and VAEs, trained across client silos using federated protocols (e.g., FedAvg), with optional privacy enhancements like differential privacy or secure aggregation Weldon et al. (2021); Behera et al. (2022); Ling et al. (2024); Little et al. (2023).

**EHR Foundation Models** Foundation models have recently gained prominence in clinical informatics, leveraging large EHR corpora to learn versatile representations for multiple tasks Vaswani & et al. (2017); Huang et al. (2019); Lee et al. (2020). Transformer-based architectures originally developed

for NLP now define the state of the art in EHR modeling Li et al. (2020b); Rasmy et al. (2021). Early efforts such as BioBERT and ClinicalBERT focused on text; GatorTron later extended transformer capacity to 8.9B parameters using over 90 billion words of clinical text Yang et al. (2022), achieving gains in concept extraction and inference tasks. Structured EHR modeling with transformers has also advanced. BEHRT Li et al. (2020b) incorporated temporality and bidirectionality to improve disease prediction. Med-BERT Rasmy et al. (2021), pretrained on structured codes from 28 million patients, achieved consistent improvements on downstream clinical classification tasks. There are great variety of models developed based on structured and unstructured data Wornow et al. (2023); Renc et al. (2024); Steinberg et al. (2023).

**Federated Timeline Synthesis** We introduce FTS, a non-trivial integration of the concepts discussed above, a federated learning framework in which clients train generative transformers on their own PHTs. Once trained generator's parameters are communicated to a central server as demonstrated in Fig. 2. At the server, this generator can on-demand synthesize customized (to achieve cohort balancing and and fairness) unlimited token sequences to train *Global Generator* (GG) without additional client interaction. By exchanging only model weights, FTS aims to achieve strong privacy guarantees without expensive cryptographic machinery, substantially reduces communication overhead compared to iterative gradient exchanges or bulk synthetic-data transfers, and does not require task-specific finetuning neither on client or server side due to zero-shot design of PHTs. The GG model can be deployed back to contributing or new clients (Fig. 2) to perform zero-shot inference, or generate synthetic PHTs for local model training.

**Significance of Federated Timeline Synthesis** By converting heterogeneous clinical records into a sequence of discrete tokens, interval tokens for time gaps, quantile tokens for continuous variables (e.g., labs and vitals), and hierarchical tokens for high-cardinality codes, PHTs offer three key advantages. First, they aim to provide strong privacy guarantees: raw timestamps and exact values remain local, and the tokenization process obscures fine-grained information before any model accesses the data. Real PHTs never leave the client. Second, they establish a common, language-like vocabulary that accommodates missingness Qian et al. (2025), irregular sampling, and inter-institutional heterogeneity in both patient populations and documentation practices. This enables transformer models to capture long-range temporal dependencies and causal event structure using the same mechanisms developed for natural language modeling. Third, PHTs enable multimodal integration by embedding clinical notes, images, genomics, and tabular EHR data into a unified representation, yielding a flexible and extensible input format for foundation models.

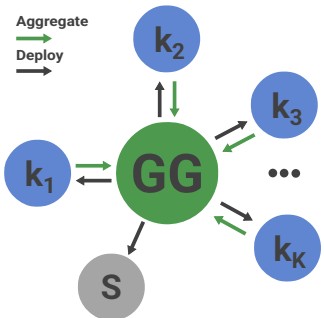

Figure 2: Federated Timeline Synthesis (FTS) workflow. Clients ($k_1$ to $k_K$) train local generative transformers on their PHTs and send trained generative models to a central server. The global generator (GG) is trained on the generated output of models. The trained GG can then be deployed to both contributing and unseen sites (S) for zero-shot inference.

When applied in a federated setting, this tokenized representation unlocks additional benefits in scalability, generalizability, and deployment flexibility. Because clients exchange only model parameters, never gradients or synthetic data, FTS significantly reduces communication overhead and eliminates the need for task-specific coordination or retraining. By task-specific coordination, we refer to the conventional requirement that participating institutions explicitly align on the details of each individual predictive task, such as defining consistent outcome variables, harmonizing label definitions, preprocessing rules, or configuring task-specific model heads. Such coordination can be burdensome, especially when institutions differ in coding practices, clinical workflows, or available data. In contrast, the FTS framework supports general-purpose foundation models trained PHTs, enabling downstream zero-shot inference across diverse tasks without requiring each site to anticipate or prepare for specific clinical endpoints. This dramatically improves scalability and makes collaborative model development more feasible in heterogeneous healthcare environments.

The globally aggregated generative model (GG) supports zero-shot inference and can be deployed across institutions regardless of language or documentation style. It can synthesize unlimited, customized token sequences or predict future PHTs for zero-shot downstream tasks. The vocabulary-driven design naturally accommodates emerging data types (e.g., new codes, medications, wearable signals, social determinants) by extending the token space while preserving backward compatibility

with previously trained models. Much like LLMs, the GG, operating over PHTs rather than free text, can serve as a base model for downstream fine-tuning or token-level augmentation, supporting a flexible and modular development path for clinical AI innovation.

**Contributions** This paper makes two primary contributions.

First, we introduce Federated Timeline Synthesis (FTS), a framework that combines generative modeling with federated learning by leveraging patient health timelines (PHTs). Instead of sharing raw data or model gradients, each institution trains a local autoregressive generator and shares only synthetic timelines, which are then aggregated into a global generator. This design reduces the need for direct data exchange and provides a communication-efficient alternative to conventional federated learning. While FTS has the potential to improve privacy, we emphasize that we do not provide formal privacy guarantees, and our approach should be viewed as complementary to, rather than a substitute for, established privacy-preserving methods such as differential privacy.

Second, we implement and empirically evaluate FTS in a controlled setting using the MIMIC-IV dataset. Specifically, we test whether models trained on synthetic PHTs generated by local client models can approximate the predictive performance of models trained directly on real data. Our experiments, focused on several clinical classification tasks, demonstrate that FTS can achieve performance close to real-data baselines under homogeneous conditions. These results support the feasibility of synthetic PHTs as a practical proxy for real data in structured clinical prediction tasks, while also highlighting open challenges around generalization to heterogeneous institutions, robustness to distributional shifts, and formal privacy analysis.

## 2 METHODS

Federated Timeline Synthesis requires medical data in tokenized timelines (PHTs) for the purpose of safety and efficiency. We describe the mathematical formalism of such approach in 2.1. We based our formulation on Renc et al. (2024). In the next section 2.2, we describe the FTS framework and provide details of implementation used in this work.

### 2.1 MEDICAL DATA REPRESENTATION, INFERENCE.

**Timeline representation:** We model each patient $p$ by a strictly ordered sequence of clinical events $\mathcal{T}_p$ as:

$$\mathcal{T}_p = \left(e_{p,1}, e_{p,2}, \ldots, e_{p,N_p}\right), \text{where} \qquad e_{p,i} = \left(\tau_{p,i}, y_{p,i}\right)$$

with *Timestamp*

$$\tau_{p,i} \in \mathbb{R}, \quad \left(\tau_{p,1}, s_{p,1}\right) <_{\text{lex}} \left(\tau_{p,2}, s_{p,2}\right) <_{\text{lex}} \cdots <_{\text{lex}} \left(\tau_{p,N_p}, s_{p,N_p}\right),$$

where $<_{\text{lex}}$ denotes lexicographic order, and $s_{p,i} \in \{1, 2, \ldots\}$ is a fixed secondary key (e.g. the index of the event's name in an alphabetically sorted dictionary) used to break ties when $\tau_{p,i} = \tau_{p,i+1}$. and *Raw payload*

$$y_{p,i} \in \bigcup_{m \in \{\text{scalar, vector, text, image,} \ldots\}} \mathcal{Y}_m \quad \text{or} \quad y_{p,i} = \emptyset,$$

where $\mathcal{Y}_m$ is the space of modality $m$ (e.g. a single lab value, a vector of vital signs, a clinical note, or an image). If $y_{p,i} = \emptyset$, that event has no payload.

At this stage, the $\mathcal{T}_p$ is simply the ordered sequence of raw events with heterogeneous payloads (or none) and captures the full, chronological clinical trajectory of patient $p$.

**Patient Health Timeline (PHT)** To convert each raw event sequence $\mathcal{T}_p$ into a discrete representation suitable for transformer input, we define a tokenization function $T$ that maps each event $e_{p,i}$ to a short subsequence of tokens:

$$T : e_{p,i} \mapsto \left(x_{p,j}, x_{p,j+1}, \ldots, x_{p,j+k_i-1}\right), \quad x_{p,j} \in \mathcal{V},$$

where $\mathcal{V}$ is the vocabulary of tokens and the index $j$ corresponds directly to the position of the first token from the event $e_{p,i}$ in the overall patient-level token sequence. Each event typically corresponds to 1–10 tokens. For example, an event occurring 6 minutes after the previous event, coded by the ICD-10 code $\texttt{E11.65}$ (Type 2 diabetes mellitus with hyperglycemia), can be tokenized as:

$$T(e_{p,i}) = \big(\underbrace{\texttt{INT}_{5\text{min}}}_{\Delta\tau}, \underbrace{\texttt{E11}}_{\text{ICD token 1}}, \underbrace{\texttt{65}}_{\text{ICD token 2}}\big).$$

Here, the 6-minute interval is rounded to the nearest predefined bin ($\texttt{INT}_{5\min}$), and the ICD code $\texttt{E11.65}$ is split into two hierarchical tokens, $\texttt{E11}$ and $\texttt{65}$, resulting in a total of $k_i = 3$ tokens. Concatenating tokenizations across all events produces the full patient-level token sequence:

$$\mathbf{x}_p = (x_{p,1},\, x_{p,2},\, \ldots,\, x_{p,L_p}),$$

and aggregating $\{\mathbf{x}_p\}_{p=1}^{P}$ over all patients provides the complete training corpus for the generative transformer. $L_p$ is the length of PHT for patient $p$.

**Types of tokens representing medical data:** The vocabulary $\mathcal{V}$ is partitioned into four token classes:

Static tokens: Patient-level attributes, time independent or slowly changing, emitted once at the start ($\tau_{p,1}$), such as age bin at the start of PHT, sex, or baseline diagnoses, marital status, socioeconomic factors. Static tokens are not optimzied in the training and always occupy start of the the timeline.

Hierarchical tokens: Multi-level categorical codes (e.g. ICD-10 "I11.65") are decomposed into successive prefixes: $\texttt{I11} \rightarrow \texttt{65}$ capturing taxonomic structure. Other examples include the Anatomical Therapeutic Chemical (ATC) classification system, which provides standardized codes for medications, indicating their therapeutic purpose and pharmacological class. Similarly, procedure codes such as CPT (Current Procedural Terminology) or ICD-10-PCS encode medical and surgical procedures performed on patients. These hierarchical coding systems enable consistent, structured representation of medications and procedures, facilitating interoperability, predictive modeling, and analysis across diverse healthcare settings.

Interval tokens: The inter-event gap $\Delta\tau_i = \tau_{p,i} - \tau_{p,i-1}$ is binned into one of $B$ nominal durations (e.g. 5 min, 1 h, 1 d, 1 w, $\ldots$), yielding $\texttt{INT}_b$. If time interval between events is shorter than some predefined threshold (typically defined as half of the shortest interval token) no time interval token is emitted.

Measurement (quantile) tokens: Each continuous measurement $v$ (e.g. a lab value or vital sign) is discretized into one of $Q$ quantiles via its empirical cumulative distribution function $F$:

$$q \;=\; \min\big(\lfloor F(v)\,Q\rfloor,\, Q-1\big), \qquad Q = 10,$$

and emitted as $\texttt{QNT}_q$. For example, a blood-pressure event $e_{p,i}$ recorded 1 minute after the previous event would yield

$$T(e_{p,i}) = \big(\; \underbrace{\texttt{BP}}_{\text{blood pressure}}, \quad \underbrace{\texttt{QNT}_5}_{\text{systolic decile}}, \quad \underbrace{\texttt{QNT}_7}_{\text{diastolic decile}} \;\big),$$

where no interval token is emitted since the 1 min gap is below the minimum 2.5-min threshold assuming 5 min is the minimum time interval token.

This tokenization preserves event order and heterogeneity, producing a unified sequence of tokens. Any further embedding (e.g. via token-type embeddings or pretrained encoders) is applied after tokenization.

**Multimodal Embeddings.** Some events carry unstructured or high-dimensional data (e.g. clinical notes, radiology images, or genomic profiles). After tokenization, each such payload $y_{p,i}^{(m)}$ is passed through a pretrained encoder:

$$\mathbf{z}_{p,i}^{(m)} \;=\; h_m\big(y_{p,i}^{(m)}\big) \;\in\; \mathbb{R}^d, \qquad m \in \{\text{notes, images, genomics}\},$$

where $h_{\text{notes}}$ is, for example, a frozen Transformer (e.g. ClinicalBERT), $h_{\text{images}}$ a frozen CNN backbone, and $h_{\text{genomics}}$ a frozen MLP. These vectors $\mathbf{z}_{p,i}^{(m)}$ are then inserted at the appropriate sequence positions.

Embedding Layer. Each discrete token $x_{p,j} \in \mathcal{V}$ is mapped to a trainable embedding via a shared lookup:

$$E : \mathcal{V} \rightarrow \mathbb{R}^d, \qquad \mathbf{e}_{p,j} = E(x_{p,j}).$$

Concatenating the token embeddings $\{\mathbf{e}_{p,j}\}_{j=1}^{L_p}$ with the frozen modality embeddings $\{\mathbf{z}_{p,i}^{(m)}\}$ in event order yields the final sequence

$$\big(\mathbf{e}_{p,1},\, \ldots,\, \mathbf{e}_{p,L_p},\, \mathbf{z}_{p,1}^{(m)},\, \ldots\big)$$

which serves as input to the transformer.

**Zero-Shot Probabilistic Inference via Future PHT Simulation** Once the global generator $f_{\theta^*}$ has been trained, and optionally fine-tuned using local data from a client not included during the initial training, we perform probabilistic inference by autoregressively sampling multiple future continuations, or *future Patient Health Timelines* (fPHTs), for each patient. Specifically, given an observed PHT prefix

$$\mathbf{x}_{p,1:L_p} = (x_{p,1}, \ldots, x_{p,L_p}),$$

we generate $N$ simulated trajectories

$$\{\tilde{\mathbf{x}}_p^{(n)}\}_{n=1}^N \sim f_{\theta^*}(\cdot \mid \mathbf{x}_{p,1:L_p}),$$

sampling tokens sequentially until a predefined stopping criterion is met (e.g., appearance of a target event token or reaching a maximum simulation horizon).

For **binary classification tasks**, consider an event $\mathcal{E}$ of interest (e.g., inpatient mortality). Let

$$M = \sum_{n=1}^N \mathbf{1}\{\mathcal{E}\text{-token} \in \tilde{\mathbf{x}}_p^{(n)}\}.$$

The probability of event $E$ is estimated as

$$\widehat{P}(\mathcal{E} \mid \mathbf{x}_{p,1:L_p}) = \frac{M}{N}.$$

For **multiclass classification tasks**, suppose the event of interest $E$ has $C$ mutually exclusive classes (e.g., discharge disposition with classes: home, rehabilitation facility, skilled nursing facility). Letting $M_c$ represent the number of trajectories ending with class $c$, we estimate the probability distribution over classes as

$$\widehat{P}(\mathcal{E} = c \mid \mathbf{x}_{p,1:L_p}) = \frac{M_c}{N}, \quad c \in \{1, \ldots, C\}, \quad \text{where} \quad \sum_{c=1}^C M_c = N.$$

For **regression tasks**, we predict continuous outcomes by extracting quantitative values from tokens generated within simulated trajectories. Let $v_n$ be the predicted quantitative value from the $n$-th simulated trajectory (e.g., lab result, vital sign measurement, time of occurrence). We estimate the regression outcome as the average:

$$\widehat{v}_p = \frac{1}{N} \sum_{n=1}^N v_n.$$

Thus, by simulating multiple fPHTs, the method produces zero-shot, scenario-based predictions that naturally account for uncertainty and temporal dependencies, flexibly accommodating binary, multiclass, and regression inference tasks in patient trajectory modeling.

## 2.2 Federated Timeline Synthesis Framework

**Training of Global Generator (GG)** We assume $K$ clients, each holding a disjoint set of tokenized Patient Health Timelines (PHTs), denoted $\text{PHT}_k$. On client $k$, we train a local autoregressive transformer generator $f_{\theta_k}$ by minimizing the standard negative log-likelihood objective:

$$\mathcal{L}_k(\theta_k) = - \sum_{p \in \text{PHT}_k} \sum_{j=1}^{L_p} \log p_{\theta_k}(x_{p,j} \mid x_{p,1:j-1}).$$

Once local training converges, each client transmits its model parameters $\{\theta_k\}$ to a central server. The server then uses these generators to produce a large synthetic corpus of pseudo-PHTs $\widetilde{\text{PHT}}$. This generation process can be guided by fixing static tokens (e.g., sex, race, or socioeconomic status) to control characteristics of the synthetic patients. Specifically:

$$\widetilde{\text{PHT}} = \bigcup_{k=1}^K \{\tilde{\mathbf{x}}_{k,i}\}_{i=1}^M, \quad \tilde{\mathbf{x}}_{k,i} \sim f_{\theta_k}(\cdot).$$

A global generator $f_{\theta*}$ is then trained on the synthetic corpus $\widetilde{\mathrm{PHT}}$ by minimizing:

$$\mathcal{L}_{\mathrm{syn}}(\theta) = - \sum_{\tilde{\mathbf{x}} \in \widetilde{\mathrm{PHT}}} \sum_{j=1}^{|\tilde{\mathbf{x}}|} \log p_\theta(\tilde{x}_j \mid \tilde{x}_{1:j-1}).$$

This two-stage process ensures that no raw or fine-grained clinical data ever leaves a client site, while the globally trained model captures aggregate patterns from all participating institutions. Once trained, the global generator $f_{\theta*}$ can be deployed back to contributing clients for local inference or fine-tuned further on real patient data from non-contributing institutions. The model can also be adapted to local needs, for example, by adding domain-specific tokens or incorporating unseen data modalities, without retraining from scratch.

## 3 EXPERIMENTS AND RESULTS

We evaluate our approach on five clinically relevant downstream tasks (DTs): DRG, SOFA score, 30-day readmission, ICU admission and in-hospital mortality prediction (see Sec. D for task definitions). All experiments are conducted on the MIMIC-IV datasetJohnson et al. (2023), which we partition at the patient level into four splits: `orig`, `test`, `val1`, and `val2`, using a 90%, 10%, 5%, and 5% ratio, respectively. Our experimental pipeline consists of four stages: (1) splitting the `orig` set into subsets, (2) selecting the optimal inference temperature for downstream task evaluation, (3) tuning the temperature for synthetic data generation, and (4) performing the final evaluation of hypothetical Federated Synthesis scenarios. Stages (1) and (2) are evaluated on `val1` and `val2` to prevent overfitting to the test set, while stages (3) and (4) are carried out on `test`.

**Experimental Details:** All models are GPT-style transformers with 3 layers, hidden dimension 768, and 12 attention heads. We use a dropout rate of 0.3 and a context window of 2048 tokens. Training is performed using the AdamW optimizer with a learning rate decaying from $6 \times 10^{-4}$ to $1 \times 10^{-5}$ over 50,000 iterations, and we train each model for 300 epochs and choose the checkpoint of the lowest loss of the last 5 validation evaluations. The effective batch size is 512. All experiments were run on nodes equipped with 8 NVIDIA A100-SXM4-40GB GPUs and 1T RAM. They training time varies across datasets from 4 to 30h.

**Training Data Division** This stage simulates a realistic scenario in which large and small healthcare facilities have access to differing volumes of electronic health record (EHR) data. For simplicity, we assume that data formats are fully harmonized across institutions.

Our goal is to identify the point at which model performance begins to degrade due to data scarcity, recognizing that the MIMIC dataset is sufficiently large for performance to plateau on a subset of data. To this end, we train models on progressively larger subsets of `orig` and evaluate them on DTs. For each setting, we compute the overall performance score across the five tasks (see Sec. E) using the `val1` split.

As shown in Tab. 3, we observe a substantial performance drop consistently across all DTs when training on 20% of the data, with further degradation at 10%. Based on these results, we define the following subsets of `orig`: `big` (80%), `small` (20%), and `little` (10%, a subset of `small`). These partitions are used in subsequent experiments to emulate institutions with varying levels of data availability.

**Inference Temperature Selection** Zero-shot inference enables the model to express uncertainty by repeatedly generating future Patient Health Timelines (fPHTs). We conduct a series of experiments varying the temperature parameter. We train a model on the `orig` split, and evaluate its performance using inference temperatures ranging from 0.7 to 1.2 on the `val2` split. Detailed results are reported in Tab. 4. Additionally, we analyze the calibration of the three best-performing temperatures on binary classification tasks in Fig. 5. The results suggest that all three achieve well-calibrated predictions. Based on the overall score and calibration curves, we find that an inference temperature of 0.9 yields the best results.

**Synthetic Data Generation Tuning** In this work, we aim to transfer knowledge from models trained on original EHR data without exposing sensitive information. This is enabled by autoregressive models, that are trained to generate data in the same format they were trained on. The knowledge transfer occurs through the generation of new PHTs, which we refer to as *synthetic*.

We hypothesize that the quality and utility of the synthetic data can be influenced by the temperature parameter used in the generation. Specifically, lower temperatures (e.g., below 1.0) may lead to more conservative generations that reflect only the most reliable patterns from the training data, potentially reducing noise. In contrast, higher temperatures (e.g., above 1.0) may introduce greater variability, potentially improving model robustness on DTs by broadening the data distribution.

To explore this, we generate synthetic versions of the `big`, `small`, and `little` splits using four temperature settings: 0.7, 0.9, 1.0, and 1.1. Each synthetic dataset is matched in patient count and demographic distribution to its original counterpart. We evaluate all generated datasets on the `test` split and compute the overall score across all DTs. For the performance evaluation, we use the temperature of 0.9 that we established in the previous experiment. Results are reported in Tab. 5, and calibration curves in Fig. 6. In addition, we perform a fidelity evaluation of the generated datasets and we report its results in C.

The results indicate that a generation temperature of 1.0 yields the best performance, and the calibration is similar across 0.9-1.1 temperature. Deviating from this default value alters the token distribution and occasionally introduces inconsistencies in the generated PHTs, such as out-of-context tokens or underrepresentation of specific token groups. A detailed analysis of token group frequencies across temperature settings is provided in Tab. 7.

Figure 3: Overall score for downstream tasks across various training datasets, including real and synthetic combinations. Each `_synth` dataset is generated to match the demographic distribution and patient count of its real counterpart.

**Evaluation of Federated Timeline Synthesis Scenarios** In the hypothetical deployment of Federated Synthesis, we consider two primary scenarios: (1) multiple institutions each contribute a generator trained on their relatively small local dataset to a central server, which aggregates them into a unified global Generator (GG); (2) a single institution utilizes the GG either directly for downstream tasks (DTs), or in combination with its own data to enhance the performance.

To evaluate these scenarios, we design experiments that simulate both contributions to and usage of the GG under varying data availability, and in various combinations with original and synthetically generated datasets. We report the results for all the setting across all DTs in Fig. 3.

The results demonstrate that synthetic data can substantially enhance model performance in low-resource settings. Notably, combining a small dataset with synthetic data generated by a model trained on a larger corpus (`small+big_synth`, `little+big_synth`) significantly boosts performance, approaching the level achieved by training directly on the `big` dataset. Moreover, aggregating synthetic data from multiple sources (`big_synth+small_synth`) outperforms using the real `small` dataset alone.

It is worth noting that `big` and `big+small_synth` achieve comparable performance, as indicated by overlapping confidence intervals. This is consistent with our earlier findings that performance on downstream tasks plateaus once the training set exceeds the size of the `small` split. However, it is also clear that knowledge is not fully preserved in the synthetic data: all models trained exclusively on synthetic datasets underperform their counterparts trained on real data (e.g., `big` vs. `big_synth`). This highlights both the potential and the current limitations of Federated Synthesis in fully capturing complex clinical patterns.

## 4 DISCUSSION AND CONCLUSION

**Summary of Results** This study introduces FTS, a novel approach to privacy-preserving foundation model training on distributed EHR data using tokenized PHTs. Our experiments across five clinically meaningful downstream tasks demonstrate that models trained on synthetic PHTs generated via FTS retain strong predictive performance. Specifically, models trained on a combination of real and synthetic data (`small+big_synth`, `little+big_synth`) perform nearly as well as those trained on the full real dataset (`big`), significantly outperforming low-resource baselines. Synthetic datasets also enable performance recovery in small data regimes and support data augmentation without sharing real patient records. While a performance gap remains between fully synthetic and fully real datasets, the gap is modest and consistent with the expected information loss in generative modeling.

**Beyond traditional classification and regression** Future PHTs support a broad spectrum of predictive, generative, and reasoning tasks in clinical AI. It extends naturally to time-to-event modeling, such as estimating the time until ICU admission or disease progression. PHTs also facilitate counterfactual reasoning, where the impact of alternative interventions can be simulated to assess potential outcomes. Through prompt conditioning and repeated sampling, models trained on PHTs can perform zero-shot clinical question answering, risk stratification, and early warning detection by identifying anomalous, high-risk patterns, or rare conditions. Additionally, embeddings extracted from PHTs can be used for patient similarity search, cohort construction, or phenotyping, uncovering latent subgroups in the population. The structured token representation also enables data imputation and missing event reconstruction, improving timeline completeness. Finally, by simulating entire cohorts, PHTs offer a path toward in silico trial design and the creation of synthetic control arms, supporting ethical and scalable clinical research without requiring access to sensitive real-world data.

**Other applications.** Although this framework is developed for healthcare time series, it naturally generalizes to other domains involving heterogeneous, sparse, and privacy-sensitive temporal data. Custom tokenization schemes would be required to adapt to specific settings, for example, in financial markets, where modeling equity price movements from stock quotes, transaction records, and proprietary signals could benefit from privacy-preserving, federated generative modeling. Potential applications include financial transaction modeling, user behavior analysis in digital platforms, industrial sensor monitoring, and longitudinal studies in social sciences. In each case, the core components of our approach, tokenized timeline representation, local generative modeling, and federated synthesis, can be adapted to enable scalable, privacy-preserving foundation model training without centralizing raw time-series data.

**Limitations** While our study introduces a novel framework for privacy-preserving model training, several limitations remain. First, we do not provide a formal privacy analysis of the proposed approach. Although federating via synthetic data generation reduces direct exposure of raw records, it does not guarantee protection against potential attacks such as membership inference or model inversion. Formal privacy-preserving mechanisms (e.g., differential privacy or secure aggregation) could be integrated with our framework, but their impact on utility and performance remains unexplored. Second, we have not demonstrated generalizability to real-world deployment scenarios, as this would require access to diverse clinical datasets and large-scale simulations across multiple institutions. Our experiments are limited to a single dataset, and generalization to heterogeneous data sources, particularly in the presence of covariate shift or institutional specialization, remains to be explored. To the best of our knowledge, MIMIC-IV is the only publicly available EHR dataset with sufficient coverage and granularity to support this type of analysis. While other datasets such as eICU Pollard et al. (2018) and AmsterdamUMCdb Thoral et al. (2021), they are restricted to the ICU setting and lack the breadth of MIMIC-IV. Third, our current framework does not incorporate multimodal information (e.g., clinical notes, imaging), which could further improve both prediction performance and the clinical realism of synthetic data. Fourth, we use a fixed model architecture across all settings to ensure consistent capacity across institutions of different sizes. This means that both small and large institutions train models with the same number of parameters, which may not be optimal. Exploring model scaling strategies relative to data availability would require extensive additional experimentation and is left for future work. Finally, while the global generator architecture provides opportunities for fairness-aware training or demographic balancing, we do not investigate such approaches in this work. In future work, we plan to address these limitations.

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

## A  Timeline Implementation for Federated Synthetic EHR Generation

Several timeline representations have been proposed for autoregressive modeling of electronic health records (EHRs). Among these, **Hierarchical Autoregressive Language mOdel (HALO)** Theodorou et al. (2023) and **Hierarchy- and Semantics-Guided Transformer (HiSGT)** Zhou & Barbieri (2025) are two notable examples that leverage hierarchical structures and semantic embeddings to improve fidelity. HALO models patient records as hierarchical timelines with visit-level and code-level granularity, but its reliance on per-visit tokenization limits its flexibility to capture fine-grained temporal patterns and to extend beyond diagnoses and selected laboratory tests. HiSGT enhances this framework by incorporating semantic information from clinical language models and taxonomies, but it similarly operates on visit-segmented sequences and lacks temporal continuity across events.

In this work, we adopt ETHOS Renc et al. (2024) as the timeline representation for our proposed Federated Synthesis (FS) framework. Unlike visit-based approaches, ETHOS represents patient records as flat, continuous sequences of tokens. This design offers several practical advantages:

**Extensibility to new data:** ETHOS removes the dependency on site-specific visit definitions, enabling the inclusion of information from outside the hospital setting, such as emergency department or outpatient visits.

**Multimodal support:** ETHOS facilitates the incorporation of diverse event types and modalities, including free-text notes and medical imaging (e.g., chest X-rays), as additional tokens within the patient timeline.

**Proven utility:** ETHOS has demonstrated good performance across a wide range of downstream tasks and currently supports parsing of nearly all structured elements in electronic medical records.

While HALO and HiSGT offer valuable design insights for centralized, visit-based synthetic EHR generation, ETHOS provides a more generalizable and extensible foundation for federated synthetic data generation. This makes ETHOS our choice for enabling large-scale, cross-institutional applications of synthetic EHR data.

## B  Computational Cost of Federated Timeline Synthesis

FTS offers notable computational efficiency compared to traditional federated learning (FL) and privacy-preserving training frameworks. Conventional FL approaches typically involve iterative gradient exchanges and frequent synchronization across clients, resulting in substantial communication and coordination overhead. In contrast, FTS requires only a one-time transmission of trained generator weights from each client, significantly reducing network traffic and simplifying orchestration.

While FTS is efficient during both the training and communication phases, its inference stage introduces additional computational overhead. Accurate prediction requires sampling multiple future Patient Health Timelines (PHTs) per patient to estimate outcome probabilities, which can be resource-intensive. However, this generative inference is performed only once per patient timeline and can support a broad range of downstream tasks, effectively amortizing the cost across multiple applications. Moreover, this overhead is mitigated by the ongoing trend of decreasing computational costs and increasing hardware efficiency.

## C  Fidelity Evaluation

To quantify the statistical alignment between real and synthetic EHR data, we adopt two recognized fidelity metrics: *Unigram Distribution* and *Dimension-Wise Correlation*. These metrics have been used in prior works Theodorou et al. (2023); Zhou & Barbieri (2025) to evaluate the preservation of marginal and patient-level code statistics. Importantly, they do not rely on visit-based tokenization, making them well-suited to our timeline-based generation framework.

**Unigram Code Distribution ($R^2$).** The Unigram score measures how well the marginal frequency of individual medical codes is preserved between real and synthetic datasets. Given the code frequency

$f_i^{\text{real}}$ in the real dataset and $f_i^{\text{synth}}$ in the synthetic dataset, the $R^2$ coefficient is computed as:

$$R^2_{\text{Unigram}} = 1 - \frac{\sum_i \left( f_i^{\text{real}} - f_i^{\text{synth}} \right)^2}{\sum_i \left( f_i^{\text{real}} - \bar{f}^{\text{real}} \right)^2} \tag{1}$$

where $\bar{f}^{\text{real}}$ is the mean frequency across all codes in the real dataset. Higher $R^2$ values indicate better alignment with the real code distribution.

**Dimension-Wise Correlation** ($R^2$). To evaluate patient-level consistency, we compute the Dimension-Wise (DimWise) correlation. For each patient $p$, we define a normalized code frequency vector:

$$\mathbf{v}_p = \frac{\text{code counts for patient } p}{\text{total codes for patient } p} \tag{2}$$

We then average these vectors across all patients in the real and synthetic datasets, obtaining $\bar{\mathbf{v}}^{\text{real}}$ and $\bar{\mathbf{v}}^{\text{synth}}$, respectively. The $R^2$ coefficient is calculated as:

$$R^2_{\text{DimWise}} = 1 - \frac{\sum_i \left( \bar{v}_i^{\text{real}} - \bar{v}_i^{\text{synth}} \right)^2}{\sum_i \left( \bar{v}_i^{\text{real}} - \bar{v}^{\text{real}} \right)^2} \tag{3}$$

where $\bar{v}^{\text{real}}$ is the mean across all dimensions in the real dataset. This metric assesses how well the overall patient-level code distributions are preserved.

**Metric Selection Justification.** While Theodorou et al. (2023); Zhou & Barbieri (2025) have included bigram and sequential bigram metrics to assess intra-visit and inter-visit code dependencies, our generation framework produces patient-level sequences without explicit visit segmentation. As a result, these visit-based metrics are not directly applicable to our evaluation setting. Moreover, if we were to treat the entire patient timeline as a single "visit" and apply bigram or sequential bigram calculations, the computational complexity would increase exponentially with sequence length, making such evaluations computationally infeasible for long patient trajectories. Therefore, we focus on *Unigram* and *DimWise* correlation, which provide scalable and meaningful, visit-agnostic assessments of statistical fidelity at both the population and patient levels.

**Implementation Details.** As described in Sec. A, our framework uses PHTs represented as flat, continuous token sequences. To ensure computational tractability during fidelity evaluation, we follow the same timeline truncation strategy used in ETHOS and limit each patient timeline to a fixed maximum length. Specifically, we compute Unigram Code Distribution and Dimension-Wise Correlation on truncated timelines capped at a predefined timeline size. Additionally, to comprehensively assess the fidelity of our synthetic data, we perform evaluations on both `timeline datasets`, representing continuous patient trajectories, and `readmission datasets`, which focus on patient episodes related to hospital readmissions. This dual evaluation provides a holistic view of the fidelity of our FTS framework across different data structures and clinical contexts.

**Evaluation Results.** Tab. 1 and Tab. 2 report the fidelity evaluation results on both the timeline and readmission datasets, respectively. Across different sampling temperatures and data scales (`big`, `small`, `little`), the model consistently achieves high Unigram and Dimension-Wise $R^2$ scores, demonstrating strong alignment with the statistical properties of real data. The results show that temperature 1.0 generally yields the best performance, achieving near-perfect correlation ($R^2 > 0.99$) across both datasets. While performance on the smaller "Little" dataset is slightly lower, especially for the readmission data where $R^2$ drops below 0.95, fidelity remains robust across all configurations. These results validate the effectiveness of our framework in generating synthetic EHR data that preserves both population-level and patient-level statistical characteristics under different sampling and data availability scenarios.

| Temperature | Big | | Small | | Little | |
|---|---|---|---|---|---|---|
| | Unigram | DimWise | Unigram | DimWise | Unigram | DimWise |
| 0.7 | 0.930 | 0.930 | 0.936 | 0.937 | 0.954 | 0.954 |
| 0.9 | 0.991 | 0.991 | 0.992 | 0.992 | 0.976 | 0.976 |
| 1.1 | 0.998 | 0.998 | 0.995 | 0.995 | 0.955 | 0.955 |
| 1.0 | 0.999 | 0.999 | 0.998 | 0.998 | 0.961 | 0.961 |

Table 1: Fidelity evaluation results on the timeline dataset across sampling temperatures (0.7, 0.9, 1.1, 1.0) and data scales (Big, Small, Little) for Unigram and Dimension-Wise $R^2$. Temperature 1.0 yields near-perfect correlation ($R^2 > 0.99$) on big and small data, while the little dataset shows slightly lower but still strong fidelity (around 0.96).

| Temperature | Big | | Small | | Little | |
|---|---|---|---|---|---|---|
| | Unigram | DimWise | Unigram | DimWise | Unigram | DimWise |
| 0.7 | 0.959 | 0.978 | 0.945 | 0.983 | 0.795 | 0.947 |
| 0.9 | 0.995 | 0.996 | 0.995 | 0.997 | 0.808 | 0.934 |
| 1.1 | 0.997 | 0.999 | 0.995 | 0.995 | 0.954 | 0.979 |
| 1.0 | 0.999 | 0.999 | 0.996 | 0.996 | 0.854 | 0.944 |

Table 2: Fidelity evaluation results on the readmission dataset across sampling temperatures (0.7, 0.9, 1.1, 1.0) and data scales (Big, Small, Little) for Unigram and Dimension-Wise $R^2$. Temperature 1.0 yields near-perfect correlation ($R^2 > 0.99$) on big, while small shows slightly lower fidelity and the little dataset has much lower Unigram (around 0.85).

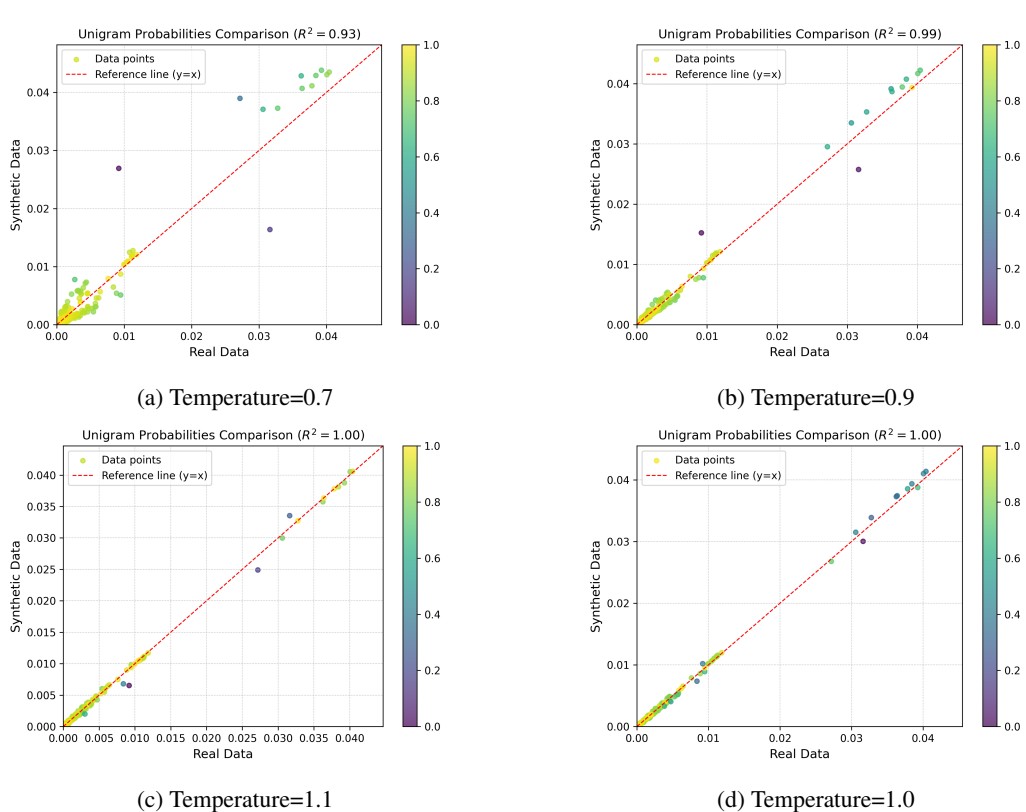

(a) Temperature=0.7

(b) Temperature=0.9

(c) Temperature=1.1

(d) Temperature=1.0

Figure 4: Comparison of unigram code distributions between real and synthetic data for the *Big* dataset under different sampling temperatures. Each subplot shows the alignment of code probabilities, with the red dashed line representing perfect agreement ($y = x$). As the temperature increases from 0.7 to 1.0, the alignment improves, reaching near-perfect correlation ($R^2 = 1.00$) at temperatures 1.0 and 1.1. This demonstrates the impact of temperature on the fidelity of the generated token distribution, with higher temperatures leading to better statistical alignment with the real data.

## D    DOWNSTREAM TASKS

We evaluate model performance across five clinically meaningful downstream tasks, encompassing classification and regression settings. All inferences and evaluations are done in zero-shot fashion.

1. **DRG Prediction (Multiclass Classification)**: The model generates a single token representing the most likely Diagnosis-Related Group (DRG) code associated with a patient's hospital stay. The prediction is made based on the entire available patient history up to the point of admission. DRG codes are used for billing and categorizing hospital cases by clinical similarity and resource usage. In the case of MIMIC-IV dataset, there are almost 800 possible DRG codes, thus, 800-class classification problem is being solved.

2. **SOFA Score Prediction (Regression)**: This task involves predicting the Sequential Organ Failure Assessment (SOFA) score, a continuous measure quantifying the extent of a patient's organ dysfunction. The model regresses the score based on historical clinical data up to the time of assessment.

3. **30-day Readmission (Binary Classification)**: The model predicts whether a patient will be readmitted to the hospital or die within 30 days of discharge. The generation starts from the last token indicating hospital discharge and continues forward in time. Both readmission and in-hospital death are treated as positive outcomes.

4. **ICU Admission (Binary Classification)**: This task predicts whether a patient will be admitted to the Intensive Care Unit (ICU) or die following a hospital admission. Generation begins from the last token corresponding to hospital admission. Both ICU admission and in-hospital death are treated as positive events.

5. **In-Hospital Mortality (Binary Classification)**: The model predicts whether a patient will die during the hospital stay. Generation starts from the last token related to hospital admission. Only death is treated as a positive label, making this a more specific and challenging binary classification task.

## E    OVERALL SCORE COMPUTATION AND CONFIDENCE INTERVALS

To provide a single, interpretable ranking across our five performance metrics, we define for each method $i$ a global score $S_i$ as an inverse-variance weighted sum of its Min–Max normalized metric values. Each metric's variance is estimated directly from its reported 95 % confidence interval, and the resulting score $S_i$ inherits an analytically derived 95 % CI. This procedure ensures that metrics with tighter uncertainty contribute more to the overall score. Details are provided below

Let $m_{i,k}$ denote the observed value of metric $k$ for method $i$, with a reported 95% confidence interval $[m_{i,k}^{\text{low}}, m_{i,k}^{\text{high}}]$. We compute a single global score $S_i$ and its 95% CI as follows.

We compute the standard error and variance for each metric:

$$h_{i,k} = \frac{m_{i,k}^{\text{high}} - m_{i,k}^{\text{low}}}{2}, \qquad \sigma_{i,k} = \frac{h_{i,k}}{1.96}, \qquad \text{Var}(m_{i,k}) = \sigma_{i,k}^2. \qquad (4)$$

Let

$$m_{(1),k} = \min_i m_{i,k}, \quad m_{(N),k} = \max_i m_{i,k}.$$

Define the normalized metric

$$\hat{m}_{i,k} = \frac{m_{i,k} - m_{(1),k}}{m_{(N),k} - m_{(1),k}} \in [0,1], \qquad (5)$$

whose variance scales as

$$\text{Var}(\hat{m}_{i,k}) = \frac{\sigma_{i,k}^2}{\left(m_{(N),k} - m_{(1),k}\right)^2}. \qquad (6)$$

The optimal weight for metric $k$ in method $i$ is

$$w_{i,k} = \frac{1/\text{Var}(\hat{m}_{i,k})}{\sum_{\ell=1}^{M} 1/\text{Var}(\hat{m}_{i,\ell})}, \quad \sum_{k=1}^{M} w_{i,k} = 1. \qquad (7)$$

The point estimate of the global score is the weighted sum

$$S_i = \sum_{k=1}^{M} w_{i,k}\, \hat{m}_{i,k}, \tag{8}$$

and under an independence assumption its variance is

$$\text{Var}(S_i) = \sum_{k=1}^{M} w_{i,k}^2 \, \text{Var}(\hat{m}_{i,k}) = \frac{1}{\sum_{k=1}^{M} 1/\text{Var}(\hat{m}_{i,k})}. \tag{9}$$

### E.1 95% CONFIDENCE INTERVAL

Finally, a 95% confidence interval for $S_i$ is

$$S_i \ \pm \ 1.96 \, \sqrt{\text{Var}(S_i)}. \tag{10}$$

## F EXTENDED RESULTS

| Data Size | DRG Classification Accuracy | SOFA Score Prediction $R^2$ | 30-day Readmission AUC | ICU Admission AUC | In-Hospital Mortality AUC | Overall Score |
|---|---|---|---|---|---|---|
| 5% | 0.235 [0.225, 0.244] | 0.458 [0.420, 0.496] | 0.716 [0.704, 0.729] | 0.868 [0.858, 0.879] | 0.848 [0.813, 0.879] | 0.000 [0.000, 0.018] |
| 10% | 0.366 [0.354, 0.376] | 0.515 [0.476, 0.550] | 0.743 [0.731, 0.755] | 0.887 [0.878, 0.896] | 0.884 [0.850, 0.911] | 0.255 [0.234, 0.275] |
| 20% | 0.511 [0.501, 0.522] | 0.542 [0.508, 0.574] | 0.758 [0.745, 0.769] | 0.901 [0.892, 0.909] | 0.886 [0.857, 0.911] | 0.531 [0.511, 0.551] |
| 30% | 0.590 [0.578, 0.601] | 0.560 [0.525, 0.593] | 0.758 [0.747, 0.770] | 0.908 [0.900, 0.917] | 0.895 [0.863, 0.917] | 0.679 [0.658, 0.700] |
| 40% | 0.655 [0.645, 0.665] | 0.575 [0.541, 0.608] | 0.766 [0.755, 0.778] | 0.909 [0.901, 0.918] | 0.908 [0.884, 0.929] | 0.803 [0.784, 0.822] |
| 50% | 0.682 [0.673, 0.693] | 0.570 [0.535, 0.604] | 0.767 [0.755, 0.778] | 0.904 [0.894, 0.912] | 0.902 [0.875, 0.926] | 0.852 [0.833, 0.870] |
| 100% | 0.761 [0.752, 0.771] | 0.578 [0.545, 0.609] | 0.775 [0.764, 0.786] | 0.907 [0.898, 0.916] | 0.901 [0.875, 0.925] | 1.000 [0.981, 1.000] |

Table 3: Performance on five downstream tasks, DRG classification, SOFA score prediction, 30-day readmission, ICU admission and in-hospital mortality, for models trained on subsets of the training data ranging from 5% to 100%. Each cell reports the mean score with its 95% confidence interval. The Overall Score column shows the aggregated performance across tasks as defined in Sec. E.

| Temperature | DRG Classification Accuracy | SOFA Score Prediction $R^2$ | 30-day Readmission AUC | ICU Admission AUC | In-Hospital Mortality AUC | Overall Score |
|---|---|---|---|---|---|---|
| 0.7 | 0.750 [0.740, 0.761] | 0.570 [0.531, 0.606] | 0.751 [0.740, 0.764] | 0.913 [0.903, 0.922] | 0.905 [0.856, 0.923] | 0.758 [0.439, 1.000] |
| 0.8 | 0.753 [0.743, 0.763] | 0.576 [0.537, 0.613] | 0.764 [0.753, 0.777] | 0.910 [0.902, 0.919] | 0.913 [0.879, 0.932] | 0.861 [0.561, 1.000] |
| 0.9 | 0.753 [0.742, 0.762] | 0.579 [0.540, 0.613] | 0.768 [0.757, 0.780] | 0.912 [0.904, 0.920] | 0.918 [0.892, 0.939] | 0.968 [0.688, 1.000] |
| 1.0 | 0.749 [0.739, 0.758] | 0.580 [0.542, 0.612] | 0.763 [0.751, 0.774] | 0.911 [0.902, 0.919] | 0.918 [0.893, 0.938] | 0.846 [0.563, 1.000] |
| 1.1 | 0.751 [0.741, 0.761] | 0.580 [0.541, 0.615] | 0.757 [0.745, 0.768] | 0.902 [0.893, 0.909] | 0.920 [0.899, 0.937] | 0.547 [0.270, 0.824] |
| 1.2 | 0.747 [0.738, 0.757] | 0.574 [0.539, 0.608] | 0.759 [0.747, 0.771] | 0.888 [0.879, 0.896] | 0.913 [0.895, 0.929] | 0.104 [0.000, 0.384] |

Table 4: Performance on five downstream tasks, DRG classification, SOFA score prediction, 30-day readmission, ICU admission and in-hospital mortality, for models evaluated using inference temperatures ranging from 0.7 to 1.2. Each cell reports the mean score with its 95% confidence interval. The Overall Score column shows the aggregated performance across tasks as defined in Sec. E.

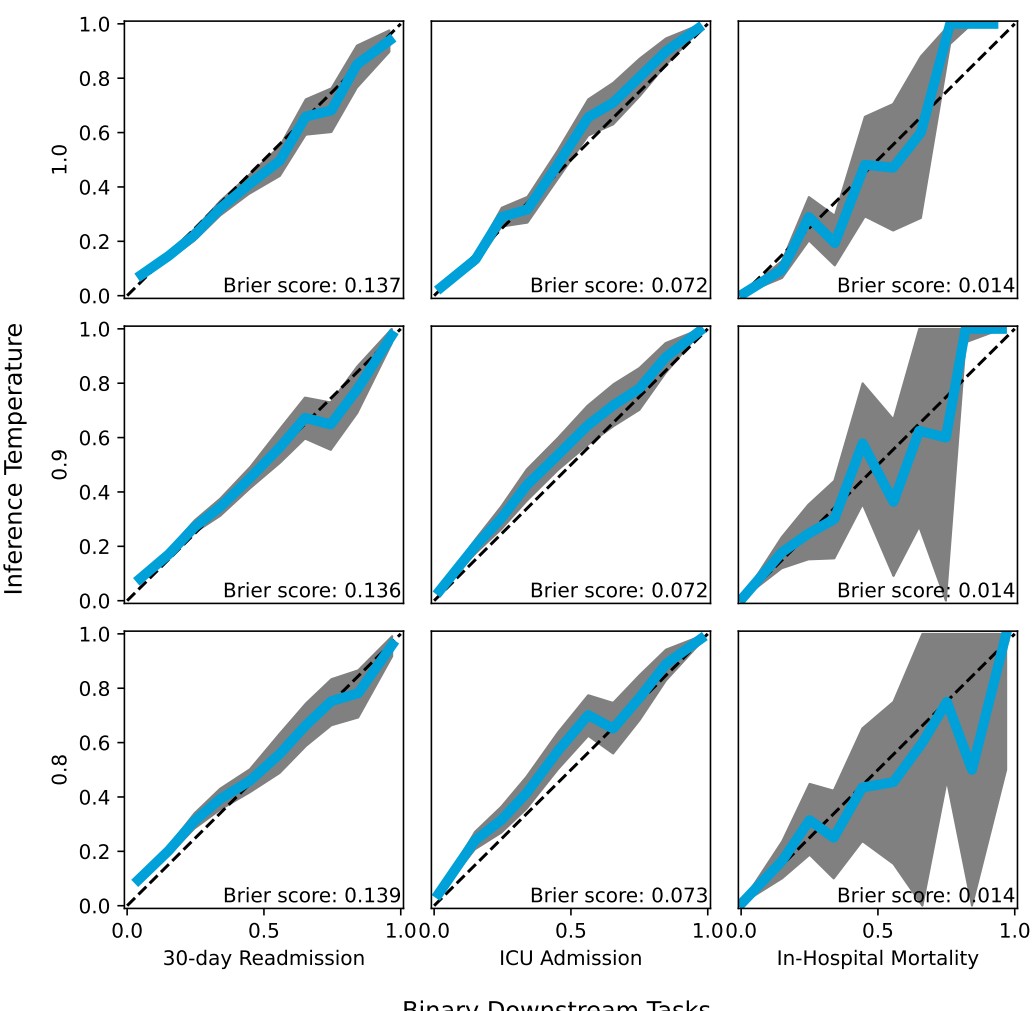

Figure 5: Calibration curves for three binary downstream tasks (30-day readmission, ICU admission, and in-hospital mortality) evaluated at inference temperatures of 1.0 (top row), 0.9 (middle row), and 0.8 (bottom row). In each panel, the solid blue line shows the observed event rate with 95% confidence bands (gray) and the dashed diagonal indicates perfect calibration. The nearly identical Brier scores across temperatures demonstrate that all temperature variants yield equally good calibration.

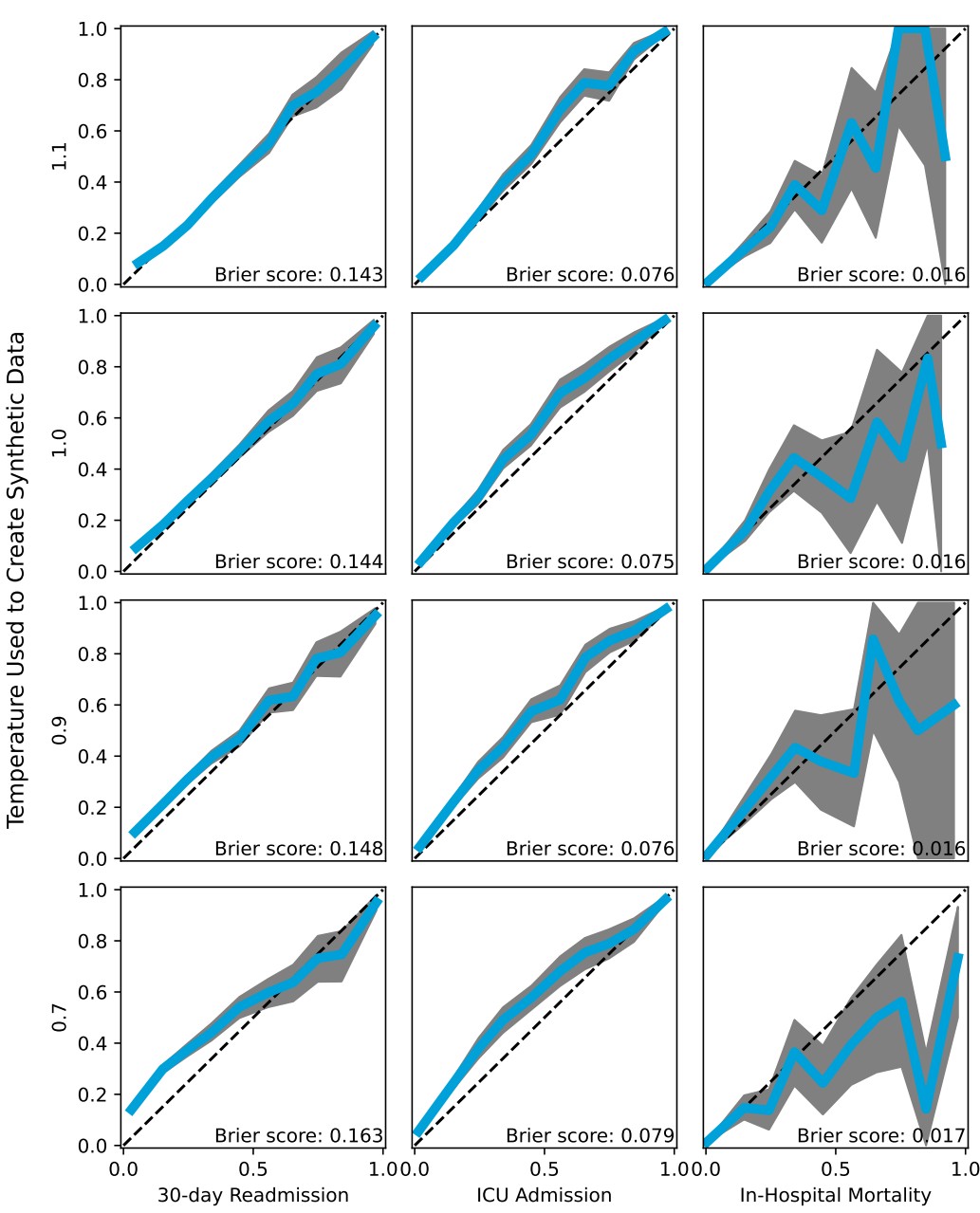

Figure 6: Calibration curves for three binary downstream tasks for model trained on data generated at temperatures of 1.1 (top row), 1.0 (second row), 0.9 (third row) and 0.7 (bottom row), and inference temperature of 0.9. In each panel, the solid blue line shows the observed event rate with 95% confidence bands (gray) and the dashed diagonal indicates perfect calibration. Brier scores are nearly identical for generation temperatures 1.1, 1.0 and 0.9, while the 0.7 setting shows worse calibration in all three tasks.

| Data | Temp. | Size | DRG Classification Accuracy | SOFA Score Prediction $R^2$ | 30-day Readmission AUC | ICU Admission AUC | In-Hospital Mortality AUC | Overall Score |
|---|---|---|---|---|---|---|---|---|
| Original | NA | big | 0.740 [0.733, 0.746] | 0.582 [0.553, 0.606] | 0.771 [0.763, 0.779] | 0.913 [0.906, 0.919] | 0.916 [0.897, 0.931] | 1.000 [0.976, 1.000] |
| | | small | 0.504 [0.496, 0.512] | 0.565 [0.538, 0.589] | 0.757 [0.750, 0.766] | 0.909 [0.903, 0.914] | 0.893 [0.873, 0.910] | 1.000 [0.976, 1.000] |
| | 1.0 | big | 0.648 [0.640, 0.655] | 0.559 [0.534, 0.585] | 0.753 [0.745, 0.762] | 0.899 [0.892, 0.905] | 0.909 [0.894, 0.924] | 0.418 [0.393, 0.444] |
| | | small | 0.366 [0.358, 0.373] | 0.532 [0.502, 0.558] | 0.725 [0.717, 0.733] | 0.883 [0.876, 0.890] | 0.873 [0.849, 0.891] | 0.418 [0.393, 0.444] |
| | 0.9 | big | 0.622 [0.614, 0.629] | 0.552 [0.524, 0.579] | 0.747 [0.739, 0.756] | 0.897 [0.891, 0.904] | 0.885 [0.848, 0.905] | 0.358 [0.332, 0.384] |
| Synthetic | | small | 0.370 [0.362, 0.377] | 0.517 [0.491, 0.544] | 0.729 [0.720, 0.737] | 0.888 [0.881, 0.895] | 0.853 [0.828, 0.880] | 0.358 [0.332, 0.384] |
| | 1.1 | big | 0.629 [0.622, 0.637] | 0.555 [0.529, 0.577] | 0.753 [0.744, 0.761] | 0.900 [0.893, 0.905] | 0.888 [0.869, 0.905] | 0.259 [0.234, 0.284] |
| | | small | 0.322 [0.315, 0.329] | 0.534 [0.509, 0.561] | 0.726 [0.718, 0.735] | 0.888 [0.881, 0.894] | 0.885 [0.862, 0.903] | 0.259 [0.234, 0.284] |
| | 0.7 | big | 0.550 [0.542, 0.558] | 0.522 [0.495, 0.548] | 0.726 [0.717, 0.735] | 0.890 [0.883, 0.897] | 0.848 [0.806, 0.875] | 0.000 [0.000, 0.026] |
| | | small | 0.303 [0.296, 0.310] | 0.487 [0.459, 0.514] | 0.709 [0.700, 0.718] | 0.872 [0.863, 0.880] | 0.835 [0.789, 0.866] | 0.000 [0.000, 0.026] |

Table 5: Performance on five downstream tasks, DRG classification, SOFA score prediction, 30-day readmission, ICU admission and in-hospital mortality. Models were trained on the original dataset or on synthetic datasets generated at inference temperatures of 1.0, 0.9, 1.1 and 0.7. For each data source and temperature, results are shown separately for `big` and `small` training sizes. We did not generate synthetic data for `little` because the model overfitted at that scale and failed to produce sensible patient timelines. Each cell reports the score with its 95% confidence interval. The Overall Score column shows the aggregated performance measure defined in Sec. E.

| Dataset | DRG Classification Accuracy | SOFA Score Prediction $R^2$ | 30-day Readmission AUC | ICU Admission AUC | In-Hospital Mortality AUC | Overall Score |
|---|---|---|---|---|---|---|
| big | 0.740 [0.733, 0.746] | 0.582 [0.555, 0.608] | 0.771 [0.763, 0.779] | 0.913 [0.907, 0.918] | 0.916 [0.896, 0.930] | 1.000 [0.987, 1.000] |
| big+small_synth | 0.729 [0.723, 0.736] | 0.573 [0.548, 0.598] | 0.763 [0.755, 0.771] | 0.913 [0.908, 0.919] | 0.911 [0.893, 0.926] | 0.978 [0.965, 0.991] |
| small+big_synth | 0.687 [0.680, 0.695] | 0.567 [0.542, 0.590] | 0.758 [0.751, 0.766] | 0.906 [0.900, 0.912] | 0.903 [0.882, 0.919] | 0.892 [0.877, 0.906] |
| little+big_synth | 0.669 [0.661, 0.676] | 0.566 [0.539, 0.590] | 0.756 [0.748, 0.764] | 0.907 [0.901, 0.912] | 0.898 [0.874, 0.916] | 0.857 [0.842, 0.871] |
| big_synth | 0.648 [0.640, 0.655] | 0.559 [0.532, 0.584] | 0.753 [0.745, 0.761] | 0.899 [0.892, 0.905] | 0.909 [0.890, 0.924] | 0.813 [0.798, 0.827] |
| big_synth+small_synth | 0.638 [0.630, 0.645] | 0.556 [0.530, 0.580] | 0.746 [0.738, 0.755] | 0.898 [0.891, 0.904] | 0.880 [0.854, 0.901] | 0.792 [0.777, 0.806] |
| small | 0.504 [0.496, 0.512] | 0.565 [0.538, 0.590] | 0.757 [0.749, 0.766] | 0.909 [0.903, 0.915] | 0.893 [0.871, 0.909] | 0.540 [0.525, 0.556] |
| little+small_synth | 0.450 [0.442, 0.458] | 0.550 [0.524, 0.577] | 0.741 [0.733, 0.750] | 0.898 [0.891, 0.904] | 0.894 [0.873, 0.912] | 0.428 [0.412, 0.443] |
| little | 0.364 [0.356, 0.371] | 0.527 [0.501, 0.552] | 0.736 [0.728, 0.744] | 0.896 [0.890, 0.902] | 0.905 [0.890, 0.918] | 0.261 [0.246, 0.276] |
| small_synth | 0.366 [0.358, 0.373] | 0.532 [0.502, 0.558] | 0.725 [0.716, 0.733] | 0.883 [0.876, 0.889] | 0.873 [0.850, 0.891] | 0.256 [0.241, 0.271] |
| little_synth | 0.240 [0.233, 0.247] | 0.485 [0.455, 0.514] | 0.710 [0.702, 0.719] | 0.857 [0.850, 0.864] | NA | 0.000 [0.000, 0.013] |

Table 6: Results on five downstream tasks for models trained on various combinations of original and synthetic datasets. Names without suffix refer to original data; names ending in _synth refer to purely synthetic data; mixed names (for example, `big+small_synth`) combine original and synthetic samples. Each cell reports the mean score with its 95% confidence interval. NA indicates that results could not be generated due to data scarcity and the fixed model size. The Overall Score column shows the aggregated performance defined in Sec. E.

| Code Group | Original Count | N | Synth_temp1 Count | N | Synth_temp0.9 Count | N | Synth_temp0.7 Count | N | Synth_temp1.1 Count | N |
|---|---|---|---|---|---|---|---|---|---|---|
| LAB | 72,174,268 | 200 | 71,106,715 | 200 | 59,794,628 | 200 | 60,518,244 | 200 | 93,198,749 | 200 |
| ATC | 20,858,757 | 87 | 22,795,722 | 83 | 12,452,636 | 83 | 6,101,998 | 77 | 37,949,275 | 86 |
| ATC_4 | 20,858,744 | 12 | 22,795,876 | 12 | 12,452,591 | 11 | 6,101,190 | 11 | 37,950,129 | 12 |
| ATC_SFX | 20,769,779 | 208 | 22,705,112 | 195 | 12,394,075 | 184 | 6,075,559 | 157 | 37,832,338 | 205 |
| Q1 | 9,494,082 | 1 | 8,861,399 | 1 | 7,572,188 | 1 | 10,245,103 | 1 | 11,804,981 | 1 |
| Q2 | 8,621,537 | 1 | 8,588,113 | 1 | 7,134,201 | 1 | 6,742,542 | 1 | 11,356,270 | 1 |
| Q3 | 8,288,665 | 1 | 8,283,252 | 1 | 6,984,193 | 1 | 6,550,402 | 1 | 10,732,378 | 1 |
| Q4 | 7,616,529 | 1 | 7,553,914 | 1 | 6,398,353 | 1 | 5,986,846 | 1 | 9,714,675 | 1 |
| Q5 | 7,601,185 | 1 | 7,594,698 | 1 | 6,539,317 | 1 | 6,199,183 | 1 | 9,592,139 | 1 |
| Q7 | 7,285,786 | 1 | 7,300,422 | 1 | 6,344,285 | 1 | 6,242,105 | 1 | 9,191,142 | 1 |
| Q6 | 7,213,924 | 1 | 7,237,419 | 1 | 6,231,882 | 1 | 5,928,606 | 1 | 9,150,938 | 1 |
| Q8 | 6,755,991 | 1 | 6,784,355 | 1 | 5,858,041 | 1 | 5,732,405 | 1 | 8,685,969 | 1 |
| Q9 | 6,510,118 | 1 | 6,482,137 | 1 | 5,702,253 | 1 | 6,048,333 | 1 | 8,233,991 | 1 |
| ICD_CM | 6,230,466 | 2,880 | 6,622,075 | 2,577 | 4,477,122 | 2,431 | 1,849,592 | 2,202 | 8,643,801 | 2,750 |
| Q10 | 5,960,105 | 1 | 5,653,703 | 1 | 5,257,956 | 1 | 7,745,200 | 1 | 7,093,716 | 1 |
| ICD_PCS | 3,197,383 | 34 | 2,942,837 | 34 | 2,083,372 | 34 | 1,078,723 | 34 | 4,133,904 | 34 |
| VITAL | 1,560,547 | 1 | 1,589,742 | 1 | 2,097,787 | 1 | 3,442,893 | 1 | 1,129,540 | 1 |
| 1h15m-2h | 1,532,311 | 1 | 1,595,616 | 1 | 953,791 | 1 | 438,114 | 1 | 2,514,137 | 1 |
| 3h-5h | 1,481,319 | 1 | 1,401,654 | 1 | 954,219 | 1 | 649,646 | 1 | 1,930,502 | 1 |
| 2h-3h | 1,468,528 | 1 | 1,467,322 | 1 | 912,352 | 1 | 455,639 | 1 | 2,186,806 | 1 |
| 15m-45m | 1,400,879 | 1 | 1,524,672 | 1 | 850,931 | 1 | 383,343 | 1 | 2,749,798 | 1 |
| BMI | 1,190,022 | 10 | 1,134,606 | 11 | 1,583,297 | 11 | 2,725,486 | 11 | 794,648 | 11 |
| 45m-1h15m | 1,147,674 | 1 | 1,218,088 | 1 | 686,810 | 1 | 280,946 | 1 | 2,086,068 | 1 |
| 5h-8h | 911,451 | 1 | 841,629 | 1 | 594,188 | 1 | 385,975 | 1 | 1,110,288 | 1 |
| 5m-15m | 907,753 | 1 | 1,026,894 | 1 | 519,283 | 1 | 210,106 | 1 | 1,989,240 | 1 |
| 8h-12h | 797,169 | 1 | 741,521 | 1 | 789,513 | 1 | 1,206,250 | 1 | 769,878 | 1 |
| TRANSFER | 599,818 | 38 | 579,025 | 38 | 409,007 | 38 | 203,179 | 38 | 818,548 | 38 |
| 12h-18h | 571,804 | 1 | 569,864 | 1 | 612,846 | 1 | 790,878 | 1 | 549,053 | 1 |
| 2mt-6mt | 367,454 | 1 | 388,899 | 1 | 469,075 | 1 | 740,404 | 1 | 324,090 | 1 |
| =6mt | 350,714 | 1 | 320,753 | 1 | 375,142 | 1 | 489,934 | 1 | 271,160 | 1 |
| 30d-2mt | 340,770 | 1 | 363,176 | 1 | 426,013 | 1 | 530,503 | 1 | 303,286 | 1 |
| INSURANCE | 310,529 | 3 | 291,693 | 3 | 233,662 | 3 | 128,742 | 3 | 332,829 | 3 |
| HOSPITAL_DISCHARGE | 310,529 | 1 | 295,194 | 1 | 237,341 | 1 | 130,433 | 1 | 325,726 | 1 |
| DISCHARGE_LOCATION | 310,529 | 10 | 295,409 | 10 | 237,408 | 10 | 130,470 | 10 | 326,145 | 10 |
| HOSPITAL_ADMISSION | 310,529 | 1 | 291,589 | 1 | 233,632 | 1 | 128,740 | 1 | 332,467 | 1 |
| DRG | 310,529 | 770 | 293,586 | 749 | 236,394 | 741 | 130,432 | 698 | 333,297 | 763 |
| ADMISSION_TYPE | 310,529 | 3 | 291,654 | 3 | 233,661 | 3 | 128,746 | 3 | 332,627 | 3 |
| 12d-20d | 309,052 | 1 | 310,314 | 1 | 359,927 | 1 | 378,336 | 1 | 261,646 | 1 |
| 20d-30d | 270,656 | 1 | 277,064 | 1 | 341,095 | 1 | 569,284 | 1 | 230,988 | 1 |
| 4d-7d | 264,533 | 1 | 255,286 | 1 | 292,271 | 1 | 492,847 | 1 | 228,224 | 1 |
| 7d-12d | 260,717 | 1 | 262,237 | 1 | 278,118 | 1 | 286,115 | 1 | 232,108 | 1 |
| 1d-2d | 242,652 | 1 | 224,712 | 1 | 201,722 | 1 | 327,574 | 1 | 245,670 | 1 |
| ED_REGISTRATION | 212,943 | 1 | 199,513 | 1 | 159,303 | 1 | 87,303 | 1 | 226,059 | 1 |
| ED_OUT | 212,943 | 1 | 201,389 | 1 | 160,945 | 1 | 88,234 | 1 | 227,490 | 1 |
| TIMELINE_END | 192,773 | 1 | 192,773 | 1 | 192,773 | 1 | 192,773 | 1 | 192,773 | 1 |
| TIMELINE_START | 192,773 | 1 | 192,985 | 1 | 192,967 | 1 | 192,967 | 1 | 193,043 | 1 |
| 2d-4d | 179,782 | 1 | 166,411 | 1 | 153,825 | 1 | 126,724 | 1 | 169,488 | 1 |
| 18h-1d | 179,474 | 1 | 172,924 | 1 | 130,567 | 1 | 69,677 | 1 | 214,290 | 1 |
| HCPCS | 101,768 | 63 | 95,482 | 40 | 68,595 | 39 | 28,201 | 27 | 120,700 | 55 |
| ICU_DISCHARGE | 52,560 | 1 | 53,052 | 1 | 32,413 | 1 | 15,864 | 1 | 96,948 | 1 |
| SOFA | 52,560 | 1 | 51,917 | 1 | 31,963 | 1 | 16,068 | 1 | 95,691 | 1 |
| ICU_ADMISSION | 52,560 | 1 | 51,877 | 1 | 31,960 | 1 | 16,049 | 1 | 95,532 | 1 |
| ICU_TYPE | 52,560 | 9 | 51,894 | 9 | 31,962 | 9 | 16,059 | 9 | 95,654 | 9 |
| MEDS_DEATH | 21,022 | 1 | 22,423 | 1 | 15,050 | 1 | 7,515 | 1 | 34,980 | 1 |
| GENDER | 0 | 0 | 21 | 2 | 12 | 2 | 3 | 1 | 46 | 2 |
| MARITAL | 0 | 0 | 16 | 5 | 6 | 3 | 3 | 2 | 77 | 5 |
| RACE | 0 | 0 | 24 | 6 | 4 | 3 | 6 | 2 | 116 | 6 |
| Total | 238,780,034 | 4,367 | 242,612,649 | 4,017 | 183,998,923 | 3,845 | 165,768,512 | 3,525 | 339,736,051 | 4,232 |

Table 7: Token counts and number of unique tokens in each code subgroup for the original dataset and for synthetic datasets generated at temperatures 1.0, 0.9, 0.7 and 1.1. For each setting, the total token count ("Count") and the corresponding unique-token count ("N") are shown side by side. Note the unexpected presence of demographic tokens such as GENDER, MARITAL and RACE in the event timelines, and the higher frequency of TIMELINE_START compared to TIMELINE_END, both of which point to glitches in the synthetic data.

