# OpenReview forum: "Federated Timeline Synthesis: Scalable and Private Methodology For Model Training and Deployment"
_ICLR.cc/2026/Conference — ICLR 2026 Conference Withdrawn Submission_

### Official Review · Reviewer_pSTX · 2025-10-28

**Soundness:** 3
**Presentation:** 2
**Contribution:** 2
**Rating:** 4
**Confidence:** 4

**Summary:**

This work proposes Federated Timeline Synthesis (FTS), a new framework that uses a federated learning (FL) for training autoregressive transformer models on distributed electronic health record (EHR) data. Each site trains a local generator on tokenized Patient Health Timelines (PHTs) and transmits model weights to a central server. The server then uses these generators to synthesize synthetic timelines and trains a Global Generator (GG) that can generate synthetic patient data for downstream tasks. The authors evaluate the method on five clinical prediction tasks using the MIMIC-IV dataset.

**Strengths:**

1. The paper combines FL with transformer models by training local transformer generators and aggregating for synthetic data generation.

2. This work proposes a multi-model integration that combines doctor notes, structured codes, images, measurements, and genomics.

3. It proposes a Zero-Shot Probabilistic Inference approach to generate a sequential synethic data. It claims this method can generate a sequence in an irreguarly-sampled way.

**Weaknesses:**

1. Unclear motivation and objective: It is difficult to discern whether the main contribution is FLframework with transformers or timeline synthesis as a generative objective.

2. Unclear core concept: The term “timeline synthesis” could be clarified. This paper does not explicility define it or properly cites it in previous works. It seems like it is a complete new concept but it lacks clear definition.

3. Inadequate dataset: This work essentailly proposes a framework which is working on cross-site datasets. However, it only tests on MIMIC4 which is sampled paients from a singel site. This paper may need more suitable dataset such as eICU, which has over 200 hospitals.

4. No privacy analysis: It does not have privacy analysis, while it claims it in the motivation.

5. Limited experiment: It has limited experiments and lack the visualization, ablation studies, and more analysis about the irregularly-sampled data generation (I assume it is core contribution). The experiments it has barely support the claim.

**Questions:**

NA

---

### Official Review · Reviewer_yGqp · 2025-11-01

**Soundness:** 1
**Presentation:** 2
**Contribution:** 1
**Rating:** 2
**Confidence:** 4

**Summary:**

* The paper proposes **Federated Timeline Synthesis (FTS)**: silos train local AR transformers on tokenized Patient Health Timelines (PHTs) and transmit either model weights or synthetic timelines to a server. The server then trains a Global Generator from synthetic sequences and performs zero-shot downstream inference by sampling future timelines.
* Experiments on **MIMIC-IV** cover DRG multi-class, SOFA regression, readmission, ICU admission, and mortality, with ablations on data size and sampling temperatures. Results show mixed real + synthetic can approach real-only baselines, while fully synthetic lags.

**Strengths:**

* Frame FL of time series generators as **federated synthesis** by sharing compact trained synthesizers instead of gradient checkpoints or synthesized data, distinct from prior works
* Provides a clear pipeline, reasonable ablations (temperature, data regimes), and fidelity checks (unigram/DimWise) that support fidelity preservation.
* Lowered communication cost of aggregating synthetic data across silos.

**Weaknesses:**

* Privacy evidence: No formal privacy guarantees or empirical audits are presented across the paper and appendix, despite privacy being the central motivation. Note that is well acknowledged that even sharing trained generator model produces privacy risks due to model inversion attack / data reconstruction attack / membership inference attack[1].
* Prior art contrast: While the paper recognized prior work on federated learning for synthesizers[2] / aggregating synthetic data rather than gradient checkpoint[3], there is limited discussion and comparison with those works , and none of them included as baseline.
* Limited evaluation: Evaluation is on Mimic-IV database and largely homogeneous; robustness to cross-institution heterogeneity (coding skews, missingness) is untested.
* Unclear inference cost: Zero-shot via Monte-Carlo sampling is computationally heavy, and the paper lacks any analysis on run time.

[1] Hayes, Jamie, et al. "Logan: Membership inference attacks against generative models." arXiv preprint arXiv:1705.07663 (2017).
[2] Rasouli, Mohammad, Tao Sun, and Ram Rajagopal. "Fedgan: Federated generative adversarial networks for distributed data." arXiv preprint arXiv:2006.07228 (2020).
[3] Goetz, Jack, and Ambuj Tewari. "Federated learning via synthetic data." arXiv preprint arXiv:2008.04489 (2020).

**Questions:**

* What attacker model do you assume, and can you provide empirical privacy audits (membership inference/model inversion/data reconstruction) or any formal guarantees?
* How does your method compare empirically to prior “federated synthesizer” (e.g., FedGAN) and “upload synthetic data” baselines (e.g., Goetz & Tewari), and why were these not included / compared?
* What is the end-to-end runtime and sampling budget for zero-shot Monte-Carlo inference, and how does performance scale with the number of samples?

---

### Official Review · Reviewer_xCSB · 2025-11-02

**Soundness:** 2
**Presentation:** 2
**Contribution:** 2
**Rating:** 2
**Confidence:** 3

**Summary:**

This paper introduces Federated Timeline Synthesis (FTS), a novel framework for training generative transformer models across distributed, electronic health records (EHRs). Each institution trains an autoregressive model locally on Patient Health Timelines (PHTs) — tokenized representations of longitudinal patient data — and sends only model weights to a central server. The server then synthesizes data from these local generators to train a Global Generator (GG), which can be used for zero-shot inference or downstream predictive modeling. Using MIMIC-IV, the authors show that models trained on synthetic PHTs perform comparably to those trained on real data across five clinical prediction tasks.

**Strengths:**

- The approach could meaningfully reduce privacy risks and regulatory barriers in federated clinical modeling, which is a major obstacle in real-world EHR applications.

- Evaluation across five clinically meaningful tasks (DRG, SOFA, readmission, ICU admission, mortality) demonstrates reasonable robustness. The inclusion of calibration and fidelity metrics (e.g., Unigram and DimWise R²) strengthens the empirical credibility.

**Weaknesses:**

- The paper does not compare with any other federated learning method as baselines. It's unclear whether the proposed method is optimal.

- The experiment is only conducted on one dataset, MIMIC. This limits generalizability to more heterogeneous or real-world multi-institutional setups. The “federated” simulation is synthetic rather than operationally federated and from different institutes.

- The experiment settings are not comprehensive - the impact of number of clients and different amount of data are not discussed. Ablation studies on different settings are not complete.

**Questions:**

See above.

---

### Official Review · Reviewer_iMxW · 2025-11-03

**Soundness:** 3
**Presentation:** 3
**Contribution:** 3
**Rating:** 2
**Confidence:** 2

**Summary:**

This paper proposes Federated Timeline Synthesis (FTS), a framework for training generative foundation models on distributed Electronic Health Record (EHR) data. The core problem it addresses is the difficulty of training large models on clinical data, which is siloed across institutions due to privacy constraints and data heterogeneity.

First, each participating institution locally converts its EHR data into "Patient Health Timelines" (PHTs), which are language-agnostic token sequences representing temporal, categorical, and continuous clinical events. Then, each institution trains a local autoregressive transformer model on its own PHTs. Each institution sends its trained model weights (not data or gradients) to a central server in a one-shot transfer. The server uses the collection of local generators to synthesize a large corpus of "pseudo-PHTs". Finally, a single "Global Generator" (GG) model is then trained from scratch on this centralized synthetic corpus.

**Strengths:**

- The paper tackles a critical and high-impact problem: enabling collaborative, large-scale model training for healthcare while respecting the severe privacy and data-siloing constraints of the field.
- The proposed two-stage synthesis framework (local generators -> central synthetic corpus -> global generator) is an interesting alternative to traditional federated learning

**Weaknesses:**

- The paper repeatedly claims "strong privacy guarantees" in the abstract , introduction , and discussion, positioning this as a primary benefit. However, the authors explicitly contradict this in their contributions, stating, "we do not provide formal privacy guarantees" , and again in the limitations: "it does not guarantee protection against potential attacks such as membership inference or model inversion". Sending trained generator weights is not formally private; these weights can contain memorized information from the local dataset. This seems misleading.
- The model trained on the little (10%) dataset "overfitted... and failed to produce sensible patient timelines". This is a gap, as "low-resource settings" are a key use case for FL.
- Table 7 reveals "glitches in the synthetic data," such as demographic tokens (GENDER, RACE) appearing multiple times. This indicates the underlying generative models still have a ways to go. Would be good for the authors to comment on this.

**Questions:**

See weaknesses

---

### Note · Authors · 2025-11-21

**Comment:**

Thank you to all reviewers for your valuable feedback and for highlighting the shortcomings of our work. We have decided to withdraw the submission in order to further improve the study.

**Withdrawal Confirmation:**

I have read and agree with the venue's withdrawal policy on behalf of myself and my co-authors.